# Impact of Drip Irrigation and Nitrogen Application on Plant Height, Leaf Area Index, and Water Use Efficiency of Summer Maize in Southern Xinjiang

**DOI:** 10.3390/plants14060956

**Published:** 2025-03-19

**Authors:** Tao Zhu, Feng Liu, Guangning Wang, Han Guo, Liang Ma

**Affiliations:** College of Hydraulic and Civil Engineering, Xinjiang Agricultural University, Urumqi 830052, China; 18692730396@163.com (T.Z.); wlmqlf1999@163.com (F.L.); 15569220976@163.com (G.W.); guohan9910@163.com (H.G.)

**Keywords:** growth models, effective accumulated temperature, maize development, growth dynamics, rapid growth stage

## Abstract

Agricultural production faces critical challenges in arid regions due to global climate change and water scarcity. Exploring optimal water and nitrogen irrigation combinations is essential to enhancing water use efficiency and crop yields. This study employs the logistic growth model to analyze the impact of varying water and nitrogen treatments on summer maize growth in southern Xinjiang. The goal is to identify an optimal irrigation strategy to enhance maize productivity, optimize water use, and ensure precise crop management. Field experiments included three irrigation levels (W1: 80% ETc, W2: 100% ETc, W3: 120% ETc) and four nitrogen rates (N0: 0 kg/ha, N1: 168 kg/ha, N2: 306.5 kg/ha, N3: 444.5 kg/ha). A logistic growth model, incorporating effective accumulated temperature, plant height, and leaf area index (LAI), quantified growth dynamics. Maximum (v_max_) and average (v_avg_) growth rates were derived, followed by regression analysis to estimate theoretical maxima and corresponding irrigation–nitrogen requirements. The logistic model provided a good approximation of maize growth dynamics. Maximum growth rates for plant height occurred at 106% ETc and 340 kg/hm² nitrogen, with an effective accumulated temperature of 319.30 °C. LAI growth rates peaked at 105% ETc and 334 kg/hm² nitrogen, with 239.75 °C during rapid growth. Optimal water–nitrogen combinations were identified, highlighting a threshold beyond which excess application becomes counterproductive. The W2N2 combination was identified as optimal, achieving a water use efficiency of 3.04 kg/m^3^. These findings offer practical guidance for optimizing agricultural practices in arid regions.

## 1. Introduction

Maize, a staple cereal crop in China, plays a crucial role in ensuring national food security due to its consistently increasing yields. As of 2021, the maize cultivation area in Xinjiang covered 1051.05 thousand hectares, accounting for 16.73% of total agricultural sown area and a substantial 47.73% of sown area dedicated to food crops [1]. However, in southern Xinjiang, irrigation efficiency remains suboptimal, with the average irrigation water utilization coefficient below 0.55, compared to the regional average of 0.575 [2]. While the effects of water and nitrogen on maize growth are well studied, their combined influence under specific climatic and soil conditions in Southern Xinjiang remains less explored [3]. In arid regions, appropriate irrigation promotes maize growth, but excessive irrigation can lead to waterlogging, resulting in root oxygen deficiency and inhibited development. Nitrogen, a vital nutrient for maize, significantly influences its growth and development [4,5]. The interplay between water and nitrogen is a critical determinant of maize productivity. The imbalanced application of irrigation and nitrogen fertilizers can diminish their synergistic effects, potentially reducing the growth potential of maize. In southern Xinjiang, maize water uses efficiency (WUE) ranges from 1.0 to 1.5 kg/m^3^. By optimizing the water–nitrogen combination, both crop productivity and WUE can be improved, alleviating water resource stress and potentially fostering sustainable agricultural practices.

Extensive research, both domestic and international, has explored the effects of water and nitrogen on maize growth and development under drip irrigation as well as the use of growth models to predict these effects. Studies have demonstrated the significant impacts of irrigation and fertilization on maize plant height (PH) and leaf area index (LAI) [6]. Water stress and nitrogen deficiency are known to inhibit foliage growth and development [7,8]. At constant irrigation levels, PH and LAI increase with higher nitrogen application rates [9]. Furthermore, the synergistic interaction between water and nitrogen significantly influences maize growth. Optimal nitrogen application significantly enhances WUE, while well-regulated irrigation improves the plant’s ability to absorb and utilize nutrients efficiently [10,11]. The logistic growth model, initially developed to describe the relationship between bacterial relative growth rates and population density [12], has since been widely applied to simulate crop growth [13]. However, model parameters vary spatially due to regional differences in rainfall and temperature [14]. Cai [15] developed logistic models based on effective accumulated temperatures and their normalized forms, which simulate dry matter accumulation in individual maize plants. Despite their utility, these models exhibit considerable spatial and temporal variability in parameters, influenced by local rainfall and temperature conditions. Therefore, it is of great significance to study the model applicability and regionalization model parameters under different years and different regions and planting conditions.

The interplay between water availability and nitrogen nutrition is crucial, as both are essential for normal plant growth and productivity [16]. An imbalance in the supply of water or nitrogen leads to less efficiency on the other side. If not managed properly, these two factors may lead to suboptimal plant growth, reduced photosynthesis, and thus lower yield [17,18]. While much research has investigated the effects of water and nitrogen on maize growth, most studies focus on macroscopic comparisons of final growth indicators. Limited attention has been paid to the dynamic patterns of various growth indicators during the reproductive phase. The application of logistic models allows for the precise simulation of crop growth and physiological indicator fluctuations across the entire growth cycle. This enables tailored adjustments of water and nutrient requirements according to specific developmental stages.

Previous research has shown that, under specific water–nitrogen combinations, maize’s growth potential is significantly enhanced, while improper water–nitrogen application in other combinations may lead to low water and fertilizer use efficiency or yield reduction, resulting in adverse outcomes in agricultural practice [19]. Moreover, optimizing water and nitrogen application not only significantly increases maize biomass accumulation but also enhances its drought tolerance, ensuring stable yield increases under water-scarce conditions [20]. Based on existing scientific knowledge, this study aims to introduce a maize logistic growth model under the condition of effective accumulated temperature changes to evaluate the effects of different water–nitrogen combinations on maize growth parameters in the southern region of Xinjiang. Through comprehensive analysis of maize growth characteristics at different developmental stages, we attempt to reveal the intrinsic relationships between crop growth, water use efficiency, and yield [21], and explore the interaction of water and nitrogen management during different growth stages. By combining precise physiological data analysis, this study seeks to propose water–nitrogen regulation strategies suitable for agricultural production in southern Xinjiang’s arid region.

Based on this, this study proposes the following scientific hypotheses to further explore the impact of water–nitrogen management on maize growth and yield: (a) Water–Nitrogen Interaction Hypothesis: The supply of water and nitrogen has a significant effect on the plant’s growth rate and physiological indicators, which can significantly promote summer maize growth while achieving high yield and efficiency. (b) Growth Model Applicability Hypothesis: The logistic crop model can effectively fit the growth status of summer maize in arid regions and provide a theoretical basis for long-term production practices. (c) Drought Stress Growth Response Threshold Hypothesis: Proper water–nitrogen application helps alleviate drought stress, but there is a threshold to its effectiveness. These hypotheses are proposed to scientifically regulate water and nitrogen supply to provide production guidance for summer maize in arid regions, verify how water–nitrogen management can maximize crop growth potential under specific drought conditions, and achieve efficient water resource use. By clarifying the specific regulatory mechanisms that promote crop growth and development, this study aims to provide more precise and efficient water–nitrogen management strategies for maize cultivation in arid areas, as well as theoretical support and data references for corresponding agronomic research.

## 2. Results

### 2.1. Parameter Calibration and Validation of the Logistic Crop Growth Model

#### 2.1.1. Parameter Calibration

A logistic growth model was developed using effective accumulated temperature during the growth period as an independent variable, replacing specific calendar dates. Maize PH and LAI were selected as the dependent variables to investigate how these indicators of summer maize growth respond to temperature changes. Using experimental data from 2018 and 2020, the model demonstrated a high level of accuracy in representing the growth patterns of maize under various water and nitrogen conditions (Table 1). The coefficient of determination (R^2^) exceeded 0.99, indicating an excellent fit and the model’s capacity to accurately describe the developmental trajectories of PH and LAI in response to accumulated temperature variations.

#### 2.1.2. Model Validation

The logistic growth model was rigorously validated for its ability to simulate LAI. The correlation coefficients were 0.998 and 0.997, while the Nash–Sutcliffe efficiency coefficients were 0.991 and 0.996. The standardized root mean square errors were 7.9% and 5.9%, respectively. These metrics confirm a strong agreement between the simulated and observed values, with the fitting points distributed evenly around the y = x line. The model effectively captured the variations in the PH and LAI of replanted maize as a function of relative effective accumulated temperature. This suggests that the characteristic parameters derived from the logistic growth model can be used to evaluate the impact of water and nitrogen on the growth dynamics of PH and LAI in replanted maize (Figure 1 and Figure 2).

### 2.2. Influence of Water and Nitrogen on PH, LAI, and Yield in Summer Maize Cultivation

#### 2.2.1. Influence of Water and Nitrogen on Key Growth Rate Parameters

Key growth rate parameters, including growth rate and relative effective accumulated temperature during the rapid growth phase, were calculated for maize PH and LAI. The results of MANOVA are presented in Table 2. Significant differences in the characteristic parameters of the PH and LAI growth models were observed across varying levels of irrigation and nitrogen application at the 0.01 significance level. The v_avg_ and v_max_ displayed significant differences among treatments involving the interaction of water and nitrogen at the 0.05 level. Meanwhile, the x_max_ and change in value (∆x) showed significant differences at the 0.01 level. The characteristic parameters of PH differed significantly between the two years at the 0.01 level. However, no significant differences were observed when considering the combined effects of irrigation amount, nitrogen application, and year. Similarly, the characteristic parameters of LAI remained consistent across the two years, indicating that the impact of water and nitrogen on PH and LAI followed a similar pattern.

In the W2N2 treatment, the v_max_ and v_avg_ for PH were recorded. In 2018, these values were 528.09 and 352.06, respectively, whereas, in 2020, they increased to 608.88 and 405.92, respectively. At the W2 irrigation level, the v_max_ and v_avg_ of PH in the N2 nitrogen treatment reached their highest levels, with values of 528.05 and 352.06 in 2018, and 608.80 and 405.92 in 2020; these phenomena indicate that the variation in effective accumulated temperature between different years may directly affect the growth and plant height of maize. These figures represented increases of 27.08%, 11.17%, and 5.32% in 2018, and 21.83%, 7.84%, and 3.74% in 2020 of the N2 treatments compared to the N0, N1, and N3 treatments. Within the N2 nitrogen application, the v_max_ and v_avg_ for PH under W2 irrigation were 13.14% and 6.10% higher in 2018, and 11.17% and 6.30% higher in 2020, compared to the W1 and W3 treatments (Figure 3).

For the LAI, v_max_ and v_avg_ under the N2 nitrogen treatment peaked at 14.189 and 9.459 in 2018, and 13.900 and 9.933 in 2020, respectively, under the W2 irrigation level. Compared to the N0, N1, and N3 treatments, these N2 treatment values were 61.97%, 22.88%, and 14.45% higher in 2018, and 54.34%, 8.40%, and 14.57% higher in 2020. Furthermore, under the N2 nitrogen level, the v_max_ and v_avg_ for LAI with W2 irrigation were 31.09% and 14.56% higher in 2018, and 20.18% and 17.26% higher in 2020, than those observed under the W1 and W3 treatments (Figure 3).

Analyzing various nitrogen gradients under the W2 irrigation level and diverse irrigation gradients under the N2 nitrogen level highlights the significant enhancement in growth rates for PH and LAI in summer maize with optimal irrigation and nitrogen application. Conversely, excessive irrigation or nitrogen application resulted in inhibitory effects. A comparative analysis of nitrogen gradients under the W1 and W3 irrigation levels, as well as irrigation gradients under the N1 and N3 nitrogen levels, revealed consistent trends, underscoring the importance of balanced irrigation and nitrogen application for optimal maize growth.

#### 2.2.2. Regression Analysis of Characteristic Parameters Incorporating Water and Nitrogen as Interactive Factors

A logistic regression analysis was performed to model the characteristic parameters of PH and LAI, incorporating irrigation and nitrogen application as dual factors. The analysis produced regression models for the v_max_, v_avg_, and ∆x (Table 3). In the regression models for v_max_ and v_avg_, the quadratic coefficients were negative, whereas the coefficients for the irrigation variable (x) and nitrogen application variable (y) were positive. This indicates a concave-down parabolic relationship for the growth rate parameters of PH and LAI in response to water and nitrogen levels. Increasing irrigation and nitrogen application positively influenced these growth rate parameters. In contrast, the regression model for ∆x expressed positive coefficients for the quadratic terms and negative coefficients of these linear terms of x and y. This pattern suggests that the duration of the rapid growth phase for both PH and LAI follows an upward-opening parabolic trajectory in response to water and nitrogen. Thus, increased irrigation and nitrogen application extended the length of the rapid growth phase. The magnitudes of the coefficients for x, y, and their interaction term (xy) followed the order x > y > xy, reflecting the hierarchical impact of irrigation, nitrogen application, and their combined interaction. This highlights the dominant role of water availability in determining crop growth in arid regions.

Using the two-factor regression model, the theoretical maximum values for v_max_ and v_avg_ of maize PH in 2018 were calculated as 512.214 and 341.476, respectively. These values were achieved under irrigation conditions of 106% ETc and a nitrogen application rate of 340 kg/ha. The corresponding rapid growth period (∆x) was 0.289, associated with an effective accumulated temperature of 319.302 °C over 26 day. In 2020, the theoretical maximum values for v_max_ and v_avg_ were 592.631 and 395.087, respectively, under the irrigation conditions of 100% ETc and a nitrogen application rate of 332 kg/ha. The corresponding rapid growth period (∆x) was 0.272, with an effective accumulated temperature of 304.83 °C over 25 d. For maize LAI, the maximum theoretical v_max_ and v_avg_ values in 2018 were 13.12 and 8.75, respectively, achieved under irrigation conditions of 105% ETc and a nitrogen application rate of 334 kg/ha. The corresponding rapid growth period (∆x) was 0.217, associated with an effective accumulated temperature of 239.75 °C over 20 d. In 2020, the maximum theoretical v_max_ and v_avg_ values for LAI were 13.35 and 8.90, respectively, under irrigation conditions of 103% ETc and a nitrogen application rate of 322 kg/ha. The corresponding rapid growth period (∆x) was 0.205, associated with an effective accumulated temperature of 229.75 °C over 19 d. A comparative analysis revealed that, for PH to enter its rapid growth phase, the effective accumulated temperature required ranged from 304 °C to 320 °C for 25–26 d. In contrast, for LAI, the effective accumulated temperature required ranged from 229 °C to 240 °C for 19–20 d. These results indicate that, under identical water and nitrogen conditions in the same region, the effective accumulated temperature and duration necessary for initiating the rapid growth phases of PH and LAI are relatively consistent. This pattern was consistently observed across both years of experimentation.

### 2.3. WUE Across Various Water and Nitrogen Treatments

Using the experimental data on yield and water consumption of summer maize in 2020, the WUE was calculated (Table 4). The W2N2 treatment achieved the highest WUE of 3.04 kg/m^3^ for summer maize, whereas the W1N0 treatment recorded the lowest WUE of 1.92 kg/m^3^. These results highlight the importance of an optimal combination of water and nitrogen in significantly enhancing WUE. Conversely, treatments with inadequate water and no or suboptimal fertilizer levels negatively impacted crop yield, thereby reducing WUE.

At the W2 irrigation level, WUE initially increased and then declined as nitrogen application rates increased, peaking at 3.04 kg/m^3^ under the N2 treatment. This maximum value was 46.06%, 24.02%, and 11.27% higher than those observed under the N0, N1, and N3 treatments, respectively. These findings emphasize the critical role of balanced water and nitrogen management in improving WUE. Under the W3 irrigation level, WUE followed a similar pattern of increasing initially, then decreasing with higher nitrogen application rates. The highest WUE of 2.31 kg/m^3^ was recorded under the N2 treatment, representing improvements of 16.85%, 6.65%, and 9.59% compared to the N0, N1, and N3 treatments, respectively. Across all irrigation conditions, moderate nitrogen application consistently resulted in higher WUE, underscoring the significant interaction effects between water and nitrogen. Both insufficient and excessive nitrogen application weakened the synergistic benefits of water and nitrogen on WUE. At the N1 nitrogen level, WUE decreased with increasing irrigation, reaching a peak of 2.51 kg/m^3^ under the W1 irrigation treatment. This maximum value exceeded the WUE of the W2 and W3 treatments by 2.20% and 15.75%, respectively. Conversely, at the N2 nitrogen level, WUE increased initially and then declined with higher irrigation levels, peaking under the W2 treatment. This maximum value was 13.81% higher than that of the W1 treatment and 31.70% greater than that of the W3 treatment. Therefore, a balanced irrigation and nitrogen management approach significantly enhances WUE. Specifically, treatments combining low nitrogen and water levels as well as moderate nitrogen and water levels showed superior WUE compared to other treatment groups. This underscores the importance of a well-optimized water-to-nitrogen ratio in maximizing WUE and improving sustainable crop management practices.

## 3. Discussion

PH and LAI are key indicators of crop yield potential. Nutrient deficiencies that limit PH and LAI can lead to reduced yields. Conversely, an imbalance in nutrient distribution within plant organs, resulting in excessive PH and LAI, may also reduce productivity, highlighting the complex relationship between plant nutrition and yield optimization [22]. In current agricultural practices, the timing of water and nitrogen application, guided by physiological or developmental indicators, often lags behind the actual growth requirements of crops. This delay compromises the timely delivery of essential resources during critical growth periods. Integrating remote sensing data with crop models through data assimilation techniques can improve the accuracy of regional crop yield estimates. This approach offers precise and timely information on crop growth stages and resource needs, helping to mitigate delays. Our preliminary research indicates that, when effective accumulated temperature reaches approximately 60% of total accumulated temperature during growth, crops such as winter wheat and summer maize undergo a rapid increase in dry matter accumulation. This effective accumulated temperature provides a reliable and precise metric for assessing the progression of crops through developmental stages [23,24]. These findings are particularly applicable to grain production in the loess soil conditions of southern Xinjiang. In this study, we established logistic growth and regression models to analyze the interactive effects of water and nitrogen on maize growth. Although similar studies on water–nitrogen optimization have been conducted in other arid regions (such as parts of North Africa, the Middle East, and the southwestern United States), this study focuses on the specific agricultural conditions in southern Xinjiang [25]. The results ranked the factors influencing maize growth as follows: irrigation > nitrogen application > interaction between water and nitrogen.

During the rapid growth phase, the growth rate and parameters related to the relative effective accumulated temperature exhibited a pattern of initially increasing and then decreasing. This “first rise and then fall” growth pattern is the result of the interaction between water and nitrogen coupling effects and crop growth stages [24]; this was strongly supported by the significant effect of the interaction between irrigation and nitrogen application measures on the PH and LAI growth model parameters in the two-year experiment (*p* < 0.01). Proper irrigation and nitrogen application can promote rapid crop growth [26], but, if the water or nitrogen supply is too much or too little, it will cause fluctuations in the growth rate; 100% ETc (W2) coupled with 306.5 kg/ha nitrogen application could exert the best coupling effect. Through our research design with field practice, by properly adjusting the irrigation time, intensity, and nitrogen fertilizer application level, the growth curve of crops can be optimized, so as to improve the development speed and final yield of crops, while the relative effective accumulated temperature displayed the opposite trend with increasing water and nitrogen inputs. Growth rates first increase and then decrease with the relative effective accumulated temperature, but this study provides a new perspective by examining the interaction of this pattern with specific environmental factors such as irrigation and N fertilizer treatments. Specifically, the timing and intensity of irrigation combined with different nitrogen fertilizer application levels can affect the rate at which this growth pattern occurs, and thus the development and yield of summer maize; this is similar to the conclusion of [27]. This provides novel insights into optimizing irrigation and nutrient management to modulate typical growth curves, highlighting the potential of water–nitrogen coupling effects in improving crop productivity.

Optimal irrigation is crucial for maintaining favorable soil temperatures in the root zone, promoting efficient nutrient uptake and enhancing crop growth. Adequate nitrogen supply supports photosynthesis, accelerates the accumulation and translocation of photosynthates, and increases PH and LAI. However, an imbalance in water and nitrogen inputs, whether insufficient or excessive, negatively impacts maize growth and physiological performance. Our regression model calculations revealed that the timing of transitions into the rapid growth phase for PH and LAI differs by only one day, with the corresponding effective accumulated temperature varies by ≤15 °C. It remains to be confirmed whether the effective accumulated temperature required for the same crop in the same region to enter the rapid growth phase of height and LAI is a constant value, and whether this metric serves as a reliable indicator of developmental stages [28]. In this study, the logistic crop growth model was used to better fit the dynamic growth process of summer maize PH and LAI. After establishing the regional model through the measured data in 2018, the prediction and verification were carried out based on the accumulated temperature data in 2020, which also showed better fitting and prediction ability. However, further studies across different crops and regions are needed for verification.

During maize’s rapid growth phase, our research found that the maximum variation in relative effective accumulated temperature across different water and nitrogen treatments spanned only 3 d. This highlights the precision of using effective accumulated temperature as a criterion for determining the crop developmental stages. Once the effective accumulated temperature threshold for the rapid growth phase is reached, it is critical to promptly supply water and nitrogen to satisfy the accelerated growth demands. This approach optimizes WUE by aligning resource application with the crop’s developmental needs. In addition, the timing and intensity of irrigation, combined with different nitrogen levels, can affect the rate at which this growth pattern occurs. Optimized irrigation and nutrient management are able to significantly modulate typical growth curves, offering potential practical applications for improving crop productivity. Xinjiang’s arid climate, characterized by high temperatures and low rainfall, exacerbates the conflict between water resource supply and demand. Agriculture consumes over 90% of the region’s total water resources, with southern Xinjiang accounting for as much as 93.1% of total social water consumption [2]. These findings highlight strategies that could contribute to improved water use efficiency in arid-region agriculture.

## 4. Materials and Methods

### 4.1. Site Description

The field experiment was carried out in the special Orchard Experimental Base of Xinjiang Agricultural University (80°14′ E, 41°16′ N, H1133 m), Wenzu County, Aksu City, Xinjiang Uygur Autonomous Region, China. The region is situated on the northwest edge of the Tarim Basin, at the southern foot of Tomur Peak in the Tianshan Mountains. It is a typical temperate continental climate, with an average annual total solar radiation of 544.115–590.156 kJ/cm^2^, a long sunshine duration of 2855–2967 h, a frost-free period of 205–219 d, an average annual precipitation of 42.4–94.4 mm, and an average annual temperature of 11.2 °C. The site experiences a typical temperate continental climate (Table 5).

The trials were conducted in bottomless measurement pits with dimensions of 3.00 × 2.20 × 2.00 m (length × width × height). Winter wheat was the preceding crop, planted with a row spacing of 0.20 m.

### 4.2. Experimental Design

The experiments were conducted in 2018 and 2020, with both sown on June 25 and harvested on October 13; the whole growth period was 111 days. The study was conducted in a test pit with dimensions of 3.00 × 2.20 × 2.00 m (length × width × height). The sides of the test pit were concreted to isolate the interactions of solutes between the experimental subplots. This study is part of the later phase of research on the relay cropping of winter wheat and summer maize, with winter wheat being the preceding crop (seed row spacing of 0.2 m). The drip irrigation lines were arranged with a spacing of 0.8 m and, to ensure uniform irrigation and minimize evaporation, the drip lines were shallowly buried at a depth of 0.05 m underground. After the winter wheat was harvested at the end of June, the wheat stubble was left at a height of 0.15 m, with the drip irrigation line layout unchanged. Summer maize was then direct-seeded in the inter-rows of the wheat stubble. To enhance the photosynthetic capacity and field ventilation of the maize crop, a wide–narrow row planting system was adopted for maize sowing. The experimental maize variety was Xinyu 9, with wide row spacing of 0.8 m and narrow row spacing of 0.4 m. At this point, one drip irrigation line satisfied the irrigation needs for two rows of maize. The plant spacing for maize was 0.25 m, with a planting density of 7.5 × 10^4^ plants/ha.

The summer maize experiments included three irrigation levels and four nitrogen fertilizer levels, with periodic irrigation beginning at the jointing stage. The experimental design comprised 12 treatments (Table 6); three replications were set for each treatment, to ensure the reliability and scientific nature of the results. The irrigation and nitrogen levels were chosen based on preliminary trials and existing agronomic recommendations for maize cultivation in arid regions. The irrigation levels were set at 80% evapotranspiration coefficient (ETc; W1), 100% ETc (W2), and 120% ETc (W3), with irrigation ceasing at the end of August. Nitrogen fertilizer treatments included N0: No nitrogen application, N1: 168.0 kg/ha of pure nitrogen, N2: 306.5 kg/ha of pure nitrogen, and N3: 444.5 kg/ha of pure nitrogen. The treatment combination of low irrigation (W1) and high nitrogen (N3) was not included; however, all other combinations were implemented. Post-sowing, basal fertilizers were applied with irrigation at the following rates: 195.0 kg/ha of elemental phosphorus, 60.0 kg/ha of elemental potassium, and 67.5 kg/ha of elemental nitrogen. Additional nitrogen fertilizer was applied via drip irrigation at the jointing, tasseling, and silking stages in a mass ratio of 2:2:1 (Table 6). Agronomic practices, including pest and disease management, inter-cultivation, and other activities, were conducted following regional high-yield field protocols. The amount of water applied during each irrigation event was based on actual crop water consumption from the preceding irrigation cycle.

In our research, we estimated the water requirements of maize by calculating *ETc* (crop evapotranspiration), starting with the estimation of *ET*_0_ (reference evapotranspiration) as shown in Figure 4. The *Kc* value was 0.99 during the jointing to silking stage, and 1.02 from the silking to filling stage. Based on these values, we then calculated *ETc* as follows:(1)$${ET}_{C}=K_{C}\times {ET}_{0}$$
where *ET_C_* represents the crop evapotranspiration, *K_C_* is the crop coefficient, and *ET_0_* is the reference evapotranspiration.

### 4.3. Sample Collection and Measurement

#### 4.3.1. Meteorological Data

HOBO weather stations were installed at the study site to record meteorological variables, including wind speed, temperature, humidity, and rainfall. The stations were positioned at a height of 2 m, with data logged every 30 min. Additionally, 20 cm evaporating dishes were used to measure the daily evaporation rate.

#### 4.3.2. PH

For each treatment group, three representative maize plants were selected from the first and second rows of the treatment plot and marked with rubber bands. PH was measured in centimeters using a ruler, from the ground surface to the top of the plant. The average height of these three plants was calculated to determine the PH for each treatment plot. During each growth period, measurements were conducted on 10 plants per treatment, and their average value was used for analysis.

#### 4.3.3. LAI

The lengths and widths of maize leaves were measured using a ruler, and the leaf area was calculated based on the empirical equation below, incorporating a coefficient value of 0.75 [19].Measurements were repeated three times for each treatment and then averaged and recorded. The formula for calculating the leaf area is given below:(2)Leaf Area=Leaf Length×Leaf Width×0.75

#### 4.3.4. Water Consumption

The water consumption of re-seeding maize was calculated by the water balance method, and the crop water consumption should be calculated by the soil moisture content in the root zone measured in the field. The water consumption of re-seeding maize at each growth stage was calculated by the water balance principle, and the calculation formula is as follows:(3)ET=ΔW+P+I+G−R−F
where *ET* is the crop water consumption; Δ*W* is the difference in soil water storage (mm); *P* is the effective rainfall (mm); *I* is the irrigation amount (mm); and *G* is the amount of groundwater replenishment (mm) to crops. The groundwater in this experiment is located above 10 m, and there is no groundwater replenishment (*G* = 0). *R* is the surface runoff (mm), and no surface runoff is generated (*R* = 0). *F* is the amount of deep leakage in the root zone (mm), the depth of the test pit is below 100 cm, and the amount of deep leakage is ignored (*F* = 0).

#### 4.3.5. Yield

The average number of kernels per ear, 100 grain weight, ear length, and yield per square meter were measured at the mature stage of re-sowing maize, and converted to yield per unit area.

#### 4.3.6. Logistic Growth Model

A logistic growth model was developed to analyze the variations in the PH and LAI of summer maize in response to temperature fluctuations. This model utilizes the effective accumulated temperature during growth as an independent variable, with maize PH and LAI as the dependent variables. The equations for the model are expressed as follows.(4)$$P_{GDDi}=\sum_{k=1}^{i}\left[{(T}_{maxk}+T_{mink})/2-T_{base}\right]$$
where P_GDDi_ represents the accumulated thermal time (°C) for the i-th day; T_maxk_ is the maximum daily temperature (°C); T_mink_ is the minimum daily temperature (°C); and T_base_ is the base temperature for maize growth, reported to be 10 °C [20].

The logistic growth equation is as follows.(5)$$Y=A/\left(1+{Be}^{-cx}\right)$$
where *A* is the upper limit of PH or LAI, *B* is the initial value parameter, and *c* is the growth rate parameter.

By deriving the first, second, and third derivatives of the logistic growth equation, the characteristic growth parameters for maize were determined. These include the accumulated thermal time at the onset of the rapid growth period (x1), duration of the rapid growth period regarding accumulated thermal time (Δx), maximum growth rate (v_max_), average growth rate (v_avg_), and accumulated thermal time at which the growth rate is highest (x_max_).

### 4.4. Statistical Analysis

Statistical analyses were conducted using SPSS Statistics 26 to perform multivariate analysis of variance (MANOVA), correlation analysis, and regression modeling. Model accuracy was assessed by comparing simulated and observed values, with graphical plots generated using Origin2019. Interaction plots were also created to analyze characteristic growth parameters and evaluate the effects of water–nitrogen interactions.

## 5. Conclusions

This study conducted a two-year field experiment in Xinjiang, China, to explore the effects of water and nitrogen dual-factor regulation on the growth characteristics and yield of double-cropped summer maize in arid regions. The effective accumulated temperature range required for maize plant height to enter the rapid growth stage is 303.28 °C to 308.19 °C; the effective accumulated temperature range for maize leaves to enter the rapid growth stage is 389.44 °C to 412.42 °C. In actual production, it is recommended to prioritize physiological or growth indicators that signal the start of the rapid growth stage, and then adjust the water and nitrogen fertilizer supply in a timely manner based on the crop’s growth needs.

Under experimental conditions, the dual-factor regression model established in this study showed that, at an ETc irrigation level of 100% to 105% and nitrogen fertilizer application of 332 to 340 kg/hm^2^, the maize PH growth rate was maximized, with a rapid growth period of 25 to 26 days. At an ETc irrigation level of 103% to 105% and nitrogen fertilizer application of 322 to 334 kg/hm^2^, the maize LAI growth rate was maximized, with a rapid growth period of 19 to 20 days. Moreover, a good coupling effect was observed between the 100% ETc irrigation treatment and a nitrogen fertilizer application of 306.5 kg/ha, which significantly promoted the growth and development of double-cropped maize, optimized water use efficiency, and achieved 3.04 kg/m^3^.

The study’s conclusions provide an optimized theoretical framework for water and nitrogen resource allocation in water-scarce arid regions and offer data support for the formulation of regional agricultural water management policies. Considering both efficient maize growth and water use efficiency, it is recommended that, in similar study areas, irrigation quotas of 103% to 106% of ETc and nitrogen fertilizer applications of 322 to 340 kg/ha be adopted for water and fertilizer management based on regional water and soil resource characteristics.

## Figures and Tables

**Figure 1 plants-14-00956-f001:**
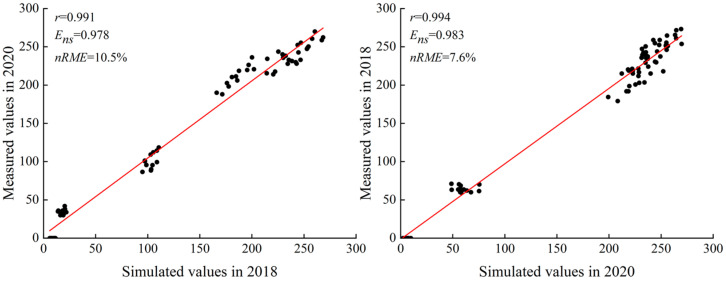
Regression comparison between simulated and measured values of logistic growth model of maize plant height.

**Figure 2 plants-14-00956-f002:**
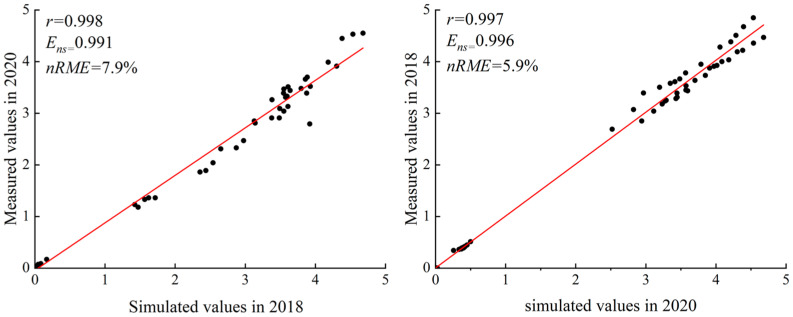
Comparison of predicted and measured LAI values for replanted maize.

**Figure 3 plants-14-00956-f003:**
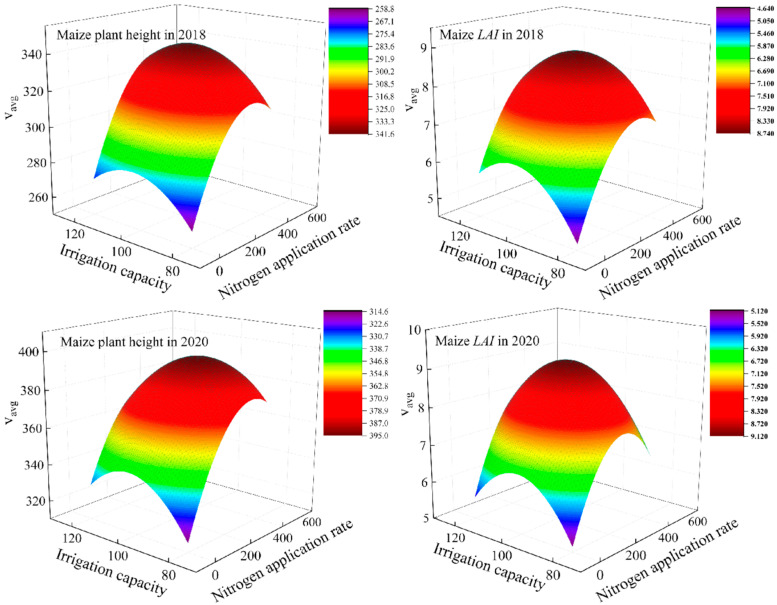
Growth rate effect of water and nitrogen interaction.

**Figure 4 plants-14-00956-f004:**
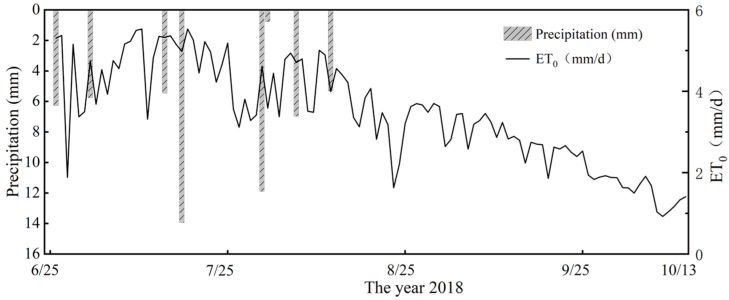
Depicts the evapotranspiration (*ET*_0_) and precipitation throughout the entire growth cycle of summer maize.

**Table 1 plants-14-00956-t001:** Logistic model parameters of maize growth under different water and nitrogen conditions.

Growth Index	Treatment	2018 Year				2020 Year			
		Curve Parameters			R^2^	Curve Parameters			R^2^
		A	B	C		A	B	C	
PH	W1N0	272.77	27.28	7.16	0.994	266.67	28.71	8.12	0.994
	W1N1	255.57	42.07	8.27	0.993	250.68	50.92	9.14	0.995
	W1N2	260.89	28.68	7.01	0.996	257.92	36.5	8.29	0.995
	W2N0	226.99	23.67	7.01	0.991	247.98	37.80	8.15	0.993
	W2N1	258.12	33.85	7.47	0.991	256.83	38.31	8.74	0.995
	W2N2	272.56	36.44	7.75	0.990	270.47	69.70	10.04	0.996
	W2N3	236.91	32.39	8.02	0.993	238.71	56.14	9.46	0.993
	W3N0	236.72	22.96	7.12	0.994	230.02	59.42	9.29	0.993
	W3N1	248.18	45.24	8.89	0.996	236.34	34.23	9.27	0.994
	W3N2	243.70	25.29	7.04	0.998	238.36	50.08	9.24	0.995
	W3N3	224.52	20.12	7.02	0.993	220.41	38.73	8.63	0.993
LAI	W1N0	4.57	193.69	10.11	0.997	4.57	318.66	10.44	0.999
	W1N1	4.4	342.49	11.26	0.998	3.44	329.04	10.56	0.997
	W1N2	4.21	271.23	10.59	0.998	4.02	415.56	10.97	0.998
	W2N0	3.45	164.98	10.23	0.998	3.44	104.18	10.24	0.998
	W2N1	4.32	390.29	11.48	0.999	3.92	653.40	12.38	0.999
	W2N2	4.71	417.79	11.05	0.997	4.49	937.37	14.91	0.998
	W2N3	3.6	445.79	12.83	0.996	4.62	288.13	9.90	0.994
	W3N0	3.29	224.02	10.65	0.995	3.26	233.38	11.05	0.998
	W3N1	3.45	389.75	12.55	0.998	3.33	608.31	13.90	0.999
	W3N2	3.59	200.95	10.48	0.996	3.54	397.31	11.54	0.996
	W3N3	2.97	249.57	10.11	0.996	2.82	317.67	11.10	0.998

**Table 2 plants-14-00956-t002:** Characteristic parameters of logistic growth model of multiple cropping maize.

Year	Factor	PH				LAI			
		x_max_	∆x	v_max_	v_avg_	x_max_	∆x	v_max_	v_avg_
2018	W3N3	0.462 ^b^	0.368 ^ab^	488.26 ^ab^	325.51 ^ab^	0.521 ^b^	0.261 ^a^	11.55 ^bc^	7.70 ^bc^
	W3N2	0.480 ^a^	0.338 ^bc^	528.39 ^a^	331.82 ^ab^	0.518 ^b^	0.234 ^c^	12.39 ^b^	8.26 ^b^
	W3N1	0.459 ^b^	0.350 ^abc^	457.21 ^bc^	317.85 ^ab^	0.529 ^b^	0.249 ^ab^	11.15 ^c^	7.43 ^bc^
	W3N0	0.439 ^cd^	0.365 ^ab^	397.80 ^d^	272.77 ^cd^	0.449 ^c^	0.258 ^ab^	8.82 ^d^	5.88 ^de^
	W2N3	0.453 ^bc^	0.339 ^bc^	482.04 ^ab^	334.27 ^ab^	0.520 ^b^	0.229 ^cd^	12.40 ^b^	8.27 ^b^
	W2N2	0.464 ^ab^	0.330 ^c^	528.09 ^a^	352.06 ^a^	0.501 ^c^	0.219 ^de^	14.19 ^a^	9.46 ^a^
	W2N1	0.434 ^cd^	0.328 ^cd^	475.00 ^abc^	316.67 ^b^	0.481 ^d^	0.205 ^f^	11.55 ^bc^	7.70 ^bc^
	W2N0	0.446 ^bcd^	0.375 ^a^	421.36 ^cd^	277.04 ^cd^	0.499 ^c^	0.247 ^b^	8.76 ^d^	5.84 ^de^
	W1N2	0.449 ^bc^	0.310 ^d^	451.58 ^bcd^	311.17 ^b^	0.475 ^d^	0.210 ^ef^	10.82 ^c^	7.22 ^c^
	W1N1	0.434 ^cd^	0.354 ^abc^	428.91 ^bcd^	302.19 ^bc^	0.541 ^a^	0.251 ^ab^	9.41 ^d^	6.27 ^d^
	W1N0	0.428 ^d^	0.375 ^a^	394.03 ^d^	262.69 ^d^	0.529 ^b^	0.261 ^a^	7.51 ^e^	5.00 ^e^
2020	W3N3	0.413 ^cde^	0.324 ^a^	541.34 ^bc^	360.89 ^bc^	0.552 ^a^	0.252 ^ab^	11.93 ^bc^	7.95 ^bc^
	W3N2	0.430 ^abc^	0.288 ^bc^	572.80 ^ab^	381.87 ^ab^	0.549 ^a^	0.249 ^abc^	11.85 ^bc^	7.90 ^bc^
	W3N1	0.434 ^ab^	0.318 ^a^	534.54 ^bc^	356.36 ^bc^	0.550 ^a^	0.240 ^bcd^	11.02 ^cd^	7.35 ^cd^
	W3N0	0.436 ^a^	0.323 ^a^	505.26 ^cd^	336.84 ^cd^	0.454 ^e^	0.257 ^a^	8.81 ^ef^	5.87 ^f^
	W2N3	0.399 ^e^	0.288 ^bc^	586.86 ^ab^	391.24 ^ab^	0.524 ^b^	0.213 ^e^	12.13 ^bc^	8.09 ^bc^
	W2N2	0.423 ^abcd^	0.262 ^c^	608.88 ^a^	405.92 ^a^	0.459^de^	0.177 ^f^	13.90 ^a^	8.55 ^b^
	W2N1	0.426 ^abcd^	0.278 ^bc^	564.55 ^ab^	376.37 ^ab^	0.468 ^d^	0.204 ^e^	12.82 ^b^	9.93 ^a^
	W2N0	0.437 ^a^	0.303 ^ab^	499.72 ^cd^	333.15 ^cd^	0.481 ^c^	0.238 ^cd^	9.01 ^e^	6.00 ^ef^
	W1N2	0.409 ^de^	0.284 ^bc^	547.72 ^bc^	365.15 ^bc^	0.461 ^de^	0.189 ^f^	11.57 ^c^	7.71 ^bcd^
	W1N1	0.399 ^e^	0.285 ^bc^	550.61 ^bc^	367.07 ^bc^	0.519 ^b^	0.228 ^d^	10.21 ^d^	6.81 ^de^
	W1N0	0.424 ^abcd^	0.305 ^ab^	475.53 ^d^	317.02 ^d^	0.519 ^b^	0.237 ^cd^	7.83 ^f^	5.22 ^f^
MANOVA	Irrigation	**	*	**	**	**	**	**	**
	Nitrogen	**	**	**	**	**	**	**	**
	Year	**	**	**	**	NS	NS	NS	NS
	Irrigation × Nitrogen	**	**	*	*	**	**	*	*
	Nitrogen × Year	NS	NS	NS	NS	**	**	NS	NS
	Irrigation × Year	NS	NS	NS	NS	*	*	NS	NS
	Irrigation × Nitrogen × Year	NS	*	NS	NS	**	**	NS	NS

Note: x_max_ represents the accumulated relative effective accumulated temperature required to reach the maximum growth rate; v_max_ represents the maximum growth rate of the replanted maize; ∆x represents the effective relative accumulated temperature corresponding to the duration of the rapid growth phase; v_avg_ represents the average growth rate of the replanted maize. Differences significant at the *p* < 0.05 and *p* < 0.01 levels are indicated by the symbols * and **, respectively; NS indicates no statistically significant difference observed in the study (*p* > 0.05); Superscript letters denote statistically significant differences between treatments (*p* < 0.05). Treatments sharing the same letter are not significantly different.

**Table 3 plants-14-00956-t003:** The regression model of logistic characteristic parameters and water and nitrogen factors.

Growth Index	Year	z	Fitted Equation	R^2^
PH	2018	v_max_	z = −220.98 − 0.0598x^2^ − 0.000781y^2^ + 0.000679xy + 12.105x + 0.460y	0.910
		v_avg_	z = −174.321 − 0.0399x^2^ − 0.000521y^2^ + 0.00453xy + 8.270x + 0.307y	0.910
		∆x	z = 0.456 + 0.00000425x^2^ + 0.000000417y^2^ + 0.00000356xy − 0.00127x − 0.000614y	0.880
	2020	v_max_	z = −270.690 − 0.0729x^2^ − 0.000732y^2^ − 0.00177xy + 15.113x + 0.663y	0.900
		v_avg_	z = −180.461 − 0.0486x^2^ − 0.000488y^2^ − 0.00118xy + 10.076x + 0.442y	0.900
		∆x	z = 0.751 + 0.0000503x^2^ + 0.000000448y^2^ − 0.00000000606xy − 0.00956x − 0.000225y	0.890
LAI	2018	v_max_	z = −30.997 − 0.00363x^2^ − 0.0000338y^2^ − 0.0000160xy + 0.765x + 0.0243y	0.975
		v_avg_	z = −20.665 − 0.00242x^2^ − 0.0000226y^2^ − 0.0000106xy + 0.510x + 0.0162y	0.975
		∆x	z = 0.727 + 0.0000463x^2^ + 0.000000406y^2^ + 0.00000227xy − 0.00947x − 0.000451y	0.910
	2020	v_max_	z = −30.094 − 0.00378x^2^ − 0.0000406y^2^ + 0.0000176xy + 0.775x + 0.0230y	0.988
		v_avg_	z = −24.838 − 0.00303x^2^ − 0.0000310y^2^ + 0.0000262xy + 0.617x + 0.0151y	0.988
		∆x	z = 0.781 + 0.0000584x^2^ + 0.000000367y^2^ + 0.00000259xy − 0.0114x − 0.000483y	0.920

**Table 4 plants-14-00956-t004:** Water use efficiency of summer maize under different water and nitrogen conditions in 2020.

Treatment	Yield (kg/ha)	Water Consumption (mm)	WUE (kg/m^3^)
W1N0	7365.72 ^h^	384.6 ^e^	1.92 ^f^
W1N1	10,092.68 ^e^	402.68 ^de^	2.51 ^bc^
W1N2	11,108.54 ^cd^	415.75 ^cde^	2.67 ^b^
W2N0	8681.14 ^g^	416.95 ^cde^	2.08 ^ef^
W2N1	10,583.63 ^de^	431.64 ^cd^	2.45 ^bcd^
W2N2	13,191.73 ^a^	433.73 ^cd^	3.04 ^a^
W2N3	12,047.86 ^b^	440.79 ^bc^	2.73 ^ab^
W3N0	9339.00 ^f^	472.68 ^ab^	1.97 ^ef^
W3N1	10,357.67 ^e^	478.42 ^a^	2.17 ^cdef^
W3N2	11,514.67 ^bc^	498.64 ^a^	2.31 ^cde^
W3N3	10,663.22 ^de^	506.03 ^a^	2.11 ^def^

Note: Superscript letters denote statistically significant differences between treatments (*p* < 0.05). Treatments sharing the same letter are not significantly different.

**Table 5 plants-14-00956-t005:** Test soil properties and initial nutrients.

Soil Layer (cm)	Ratio of Soil Particle Size (%)			Bulk Density	Field Water Holding Capacity (%)	Organic Matter (g/kg)	Available Nitrogen (mg/kg)	Available Phosphorus (mg/kg)	Quick-Acting Potassium (mg/kg)
	>0.05 mm	0.05–0.002 mm	<0.002 mm	(g/cm^3^)					
0–20	43.59	52.93	3.49	1.50	22.90	8.97	36.64	13.98	115.62
21–40	42.21	53.76	4.04	1.50	20.78	6.28	29.61	6.62	105.74
41–60	33.42	61.94	4.64	1.45	24.99	4.71	21.45	5.72	101.81

Note: Field water holding rate, mass water content rate.

**Table 6 plants-14-00956-t006:** Experimental scheme of water and nitrogen for multiple sowing maize.

Treatment		Irrigation Level (mm)	Amount of Irrigation (mm)		Bottom Fertilizer (kg/ha)	Topdressing Fertilizer (kg/ha)			Total Nitrogen Fertilizer
			2018	2020		Elongation	Tassel Emergence	Silking	
W1	N0	80% ETc	489.60	497.70	0.0	0.0	0.0	0.0	0
	NI	80% ETc	489.60	497.70	67.5	40.2	40.2	20.1	168.0
	N2	80% ETc	489.60	497.70	67.5	95.6	95.6	47.8	306.5
W2	N0	100% ETc	578.25	584.35	0.0	0.0	0.0	0.0	0
	NI	100% ETc	578.25	584.35	67.5	40.2	40.2	20.1	168.0
	N2	100% ETc	578.25	584.35	67.5	95.6	95.6	47.8	306.5
	N3	100% ETc	578.25	584.35	67.5	150.8	150.8	75.4	444.5
W3	N0	120% ETc	666.90	675.60	0.0	0.0	0.0	0.0	0
	NI	120% ETc	666.90	675.60	67.5	40.2	40.2	20.1	168.0
	N2	120% ETc	666.90	675.60	67.5	95.6	95.6	47.8	306.5
	N3	120% ETc	666.90	675.60	67.5	150.8	150.8	75.4	444.5

Note: ETc denotes the crop evapotranspiration for maize, with W representing the volume of irrigation water applied and N indicating the amount of nitrogen fertilizer used.

## Data Availability

The original contributions presented in the study are included in the article material; further inquiries can be directed to the corresponding authors.

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
