# Peer review of "Impact of Drip Irrigation and Nitrogen Application on Plant Height, Leaf Area Index, and Water Use Efficiency of Summer Maize in Southern Xinjiang"

_plants, 2025, doi:10.3390/plants14060956_

Round 1
Reviewer 1 Report
Comments and Suggestions for Authors
See the attached file

Author Response
请参阅附件

Reviewer 2 Report
Comments and Suggestions for Authors
Review: ‘Impact of drip irrigation and nitrogen application on plant height, leaf area Index, and water use efficiency of summer maize in Southern Xinjiang
General comments
In the present manuscript the Authors analyse impact of varying water and nitrogen treatments on maize plant growth. Field experiments included three irrigation levels: 80%, 100%, and120% ETc and four nitrogen rates: 0 kg/hm2 , 19 168 kg/hm2 , 306.5 kg/hm2, 444.5 kg/hm2.
Very simple assumptions of architecture were combined with growth rates to produce a theory of biological growth which has excellent predictive power of the maximum scale of organisms such as plants.
The subject of the paper fits the aim of the Journal and results could be interest for the scientific community. However, in my opinion the present paper does not meet the scientific standard for the Journal for issue of main concerns.
The manuscript is unclear and the reader does not have a clear idea of the experimental design. In the section on M&M, the description of the experimental setup is unclear, which is obvious in the results.
What is ‘summer maize’? Do the authors know about ‘winter maize’?
In my opinion, the nitrogen fertilization rates has been assumed to be too high. The rates are not adjusted to the growth stages of the plants maize. There is no detailed description of the weather conditions. In effect very difficult take a sense this experiment.
Detailed comment
- How much water was used for irrigation?
- Water consumption - how was the water consumption of plants calculated?
- What kind of yield are the authors writing about, yield corn or biomass?
- No experimental plan, no experimental plot area, no number of repetitions, no precise description of how the experiment was conducted. No experimental clearly factors: what is factor A and factor B.
- Line 306: Was corn sown on June 25th and harvested on October 13th in each year?
- Line 311: Did winter wheat was the preceding crop in each year?
- Line 313 Drip irrigation tapes were shallowly buried at a depth of approximately 5 cm. What was depth sowing of maize? How corn was sown at the recommended depth of 5-6 cm? It is not logical.
- Line 311: Winter wheat was the preceding crop, in which year ?
- Line: 332. ETc coefficient was calculated incorrectly. Determining water needs should be divided into 3 stages:
- I – Estimation of ETo indicator evapotranspiration
- II – Estimation of evapotranspiration of a specific ETR plant species
- III – Estimation of evapotranspiration of a specific ETR planting
- Figure 4. Please provide English text.
- Line 339: Please provide aggregate data for these parameters: wind speed, temperature, humidity, and rainfall.
- Line 351: How were leaf measurements taken: how many plants, how many leaves, or all of them on the plant?
- Table 6. The dates are not clearly shown in the tables. The N doses should also be total too.
- Formula 1 . lack description ETC = KC × ET
The introduction section is very weak. Authors did not show what the novelty and specific findings which can attract the readers.
Lack of properly conducted statistical analysis. Lack of examination of the normality of the distribution of variables.
The results of the research are unreliable because there is no basis for statistical analysis.
Line 239: The discussion in confused and difficult to read.
A major overhaul is needed. In this form, manuscript is unacceptable.
Comments on the Quality of English LanguageGrammatical errors in sentences.
Author Response
请参阅附件

Reviewer 3 Report
Comments and Suggestions for Authors
The authors of the article, both in the abstract and throughout the text, discuss practical recommendations for enhancing agricultural practices in arid regions. However, it raises a question regarding their reference to specific water levels, such as W1: 80% ETc, W2: 100% ETc, and W3: 120% ETc. How can these levels be effectively applied in their guidance?
In fact, they conclude that the combination W2N2 is optimal is the dynamic scenario for the arid areas, where W2 means 100% ETc.
No
Author Response
请参阅附件

Round 2
Reviewer 1 Report
Comments and Suggestions for Authors
The revised version of the manuscript is significantly improved and effectively incorporates the proposed changes. I appreciate your attention to detail and your commitment to the quality of the work.
Author Response
感谢您对修改论文的帮助
Reviewer 2 Report
Comments and Suggestions for Authors
I have no comments for the Authors. My comments have been taken into account.
Author Response
感谢您对修改论文的帮助